# Health aspects and lifestyle of licensed manual therapists during the COVID-19 pandemic in Sweden: The CAMP cohort study

Iben Axén[1,2☯], Nathan Weiss[1,3☯*], Eva Skillgate[1,3,4]

1 Unit of Intervention and Implementation Research for Worker Health, Institute of Environmental Medicine, Karolinska Institutet, Stockholm, Sweden, 2 Et liv i bevegelse (ELIB), Oslo, Norway, 3 Department of Health Promotion Science, Musculoskeletal & Sports Injury Epidemiology Center, Sophiahemmet University, Stockholm, Sweden, 4 Naprapathögskolan-Scandinavian College of Naprapathic Manual Medicine, Stockholm, Sweden

☯ These authors contributed equally to this work.
* Nathan.weiss@shh.se

## Abstract

### Background

This study assessed health and change in lifestyle factors in Swedish manual therapists during one year of the COVID-19 pandemic, and potential differences with regards to age, sex, and business constellation. Further, therapists' strategies for health promotion during the pandemic were explored.

### Methods

In this cohort study, 816 clinically active manual therapists were followed with web-based surveys during a year of the COVID-19 pandemic in Sweden, measuring physical activity, sedentary time, COVID-19-related worries, maladaptive coping, alcohol and tobacco consumption. Health promotion and impact of the pandemic on physical and mental health were explored in free text questions. Generalized estimating equations were conducted to assess changes in sample averages over time, and qualitative content analysis was used to code and categorize free-text answers.

### Results

There was a decrease in physical activity and sedentary time increased as well as subjective mental health impact by the pandemic over one year. Maladaptive coping decreased during follow-up, and alcohol and tobacco consumption decreased in younger participants, and women, respectively. Participants stated that the pandemic affected their physical and mental health and reported using health promoting activities primarily targeting physical activity, nutrition, and sleep.

**Data availability statement:** Due to ethical restrictions of disclosing personal and sensitive data in accordance with the protocol approved by the Swedish Ethical Review Authority, authors have to seek permission to allow us to make the data used in this study available. Data will be available upon request after permission is granted from the Swedish Ethical Review Authority. Inquiries for data access should be sent to the Ethical Review Authority, whose contact is registrator@etikprovning.se for permission to openly share the data.

**Funding:** This research was supported by: IA: AFA Insurance grant number 200140 https://www.afaforsakring.se/ NW: The Swedish Naprapathic Association (no grant number). https://naprapater.se/. The funders did not play any role in the study design, data collection and analysis, decision to publish, or preparation of the manuscript.

**Competing interests:** The authors have declared that no competing interests exist.

## Conclusion

Swedish manual therapists maintained good lifestyle habits except for a small decrease in physical activity and slight increase in sedentary behavior and subjective mental health impact by the pandemic over time. There were small differences in terms of maladaptive coping, alcohol consumption, and tobacco consumption, however, these differences were not likely clinically relevant. The therapists seemed conscientious regarding health promotion measures during one year of the COVID-19 pandemic.

## Introduction

The outbreak of the COVID-19 pandemic in 2020 [1] led to unprecedented public health consequences all over the world, not only in terms of disease and death, and the severe impact on health care systems, but also in the way other aspects of peoples' lives were affected. To stop viral spread, many countries enforced lockdowns, where nothing but essential societal functions were running. Some changes were that we learned to work and study from home, to avoid social interaction outside the family, to shop online, and we stopped traveling.

Sweden chose a different route: instead of lockdowns, personal responsibility in minimizing viral spread was emphasized [2]. The authorities asked for social distancing, hand hygiene and self-isolation when experiencing symptoms. Recommendations included working from home and avoiding public gatherings. It was hypothesized that voluntary compliance would be more sustainable than controlling strategies [3]. If this different route was successful or not is not known, but compared to many European countries, Sweden's excess mortality rate turned out lower between 2020 and 2021 [4]. Sweden's vaccination program against COVID-19 were initiated in the end of 2020 and was implemented in stages depending on risk status. Elderly people and frontline healthcare workers were among the first groups that received the vaccine, and thereafter it was progressively made available for other groups in the population based on age and pre-existing conditions. By November 2021, 85% of the adult population in Sweden had received their first dose [5].

As part of Sweden's COVID-19 strategy, licensed manual therapists were allowed to stay in business during the pandemic in Sweden. Manual therapists, licensed chiropractors and naprapaths, treat and prevent musculoskeletal conditions, which are prevalent conditions worldwide [6]. However, manual therapists mainly work with their hands in close proximity to the patient, which may led to a challenging work environment during the pandemic; having to balance patients' needs with infection control measures, such as social distancing.

Previous studies have examined the impact of lockdowns on people's health due to concerns that being confined to the home would lead to a more sedentary lifestyle and possibly poorer lifestyle choices [7]. Financial instability, fear of infection and

the unpredictable situation led to poor mental health in some countries [8], while in Sweden, measures of mental health remained stable over time in a general population sample of adults [9].

The physical and mental health of healthcare personnel, particularly those dealing with COVID-infected patients, have been investigated. In qualitative studies, fear was a common experience [10] but also coping with the situation and adapting to change [11,12]. Manual therapists' coping responses to the pandemic have been studied in countries of lockdown [13,14]. These studies described measures taken by manual therapists to provide safe care to patients under the circumstances. One study of US chiropractors reported high levels of stress [15]. However, studies from the Swedish context are missing.

The aim of this study was to assess health and changes in lifestyle in manual therapists during one year of the COVID-19 pandemic in Sweden, and potential differences with regards to age, sex, and business constellation. Further, the aim was to explore therapists' strategies to promote their own health during the COVID-19 pandemic.

## Materials and methods

### Study design

This study was of mixed methods design and based on the Corona And Manual Professions (CAMP) cohort study, ClinicalTrials register identifier: NCT04834583. The study was approved by the Swedish Ethical Review Authority (Dnr 2020–03836).

Extensive details regarding the recruitment and data collection procedures of the CAMP cohort study have been published previously [16] and are described briefly below.

### Participants

Clinically active chiropractors and naprapaths, licensed by the National Board of Health and Welfare in Sweden, and those undergoing licensing practice were invited to participate, and 816 manual therapists were included.

### Data collection

Participants were recruited from November 1st 2020 through January 1st, and were thereafter followed prospectively during a one-year period (at 3-, 6-, and 12-months).

Informed consent was provided by all participants on the first page of the web-based baseline survey. Before being able to answer the baseline survey, participants were presented with information about the study and had to answer yes to the following statement: "I have understood what participation in this research study means, and consent to participating".

### Variables

**Physical activity.** Physical activity (PA) was assessed with two questions adopted form the Swedish national public health survey (NBHWA-PA questions) [17,18] at baseline and at the 12-month follow-up. Participants were asked how many minutes they committed to physical exercise in a normal week (e.g., running, fitness training, ball sports), and everyday PA (e.g., walking, cycling, gardening) with the following answer alternatives: "0 minutes/no time", "1-30 minutes", "30-60 minutes", "60-90 minutes", "90-120 minutes", and "More than 120 minutes". The categorical response alternatives were converted using the middle value in each category, e.g., 30–60 minutes were converted to 45 minutes. Further, each participant's 'total physical activity' per week was calculated by summarizing their physical exercise multiplied with 2 (to account for the higher intensity of PA), and their minutes of everyday PA, a measure which has previously shown moderate concurrent validity with accelerometer PA [19].

**Sedentary time.** Sedentary time was assessed at baseline and at 12-month of follow-up with SED-GIH, a single item question concerning the number of hours per day participants were sedentary, aside from sleeping, with the following

answer alternatives: "Most of the day", "13-15 hours", "10-12 hours", "7-9 hours", "4-6 hours", and "1-3 hours" [20]. The categorical response alternatives were converted using the middle value in each category, e.g., 7–9 hours equated to 8 hours of sedentary time per day. The SED-GIH has previously displayed acceptable convergent validity [17].

**Impaired sleep.** To assess impaired sleep, the following questions were used at baseline and at 12 months of follow-up: "Do you have difficulties falling asleep?", "Do you wake up several times during the night and have difficulties falling asleep again?", and "Do you feel very tired during worktime/daily activities?". The answer alternatives were: "never", "rarely/a few times per year", "a few times per month", "several times a week", "always/every day". The first two questions were adopted from the Karolinska Sleep Questionnaire [21], and the last from the unwinding and recovery questions by Aronsson et al. [22]. Impaired sleep was defined as having difficulty initiating sleep and/or difficulty maintaining sleep accompanied by daytime consequences "several times a week" or "always/every day" [23].

**Tobacco and alcohol use.** Tobacco consumption was measured by asking the participants the following questions at baseline and 12 months of follow-up: "Do you smoke daily?" and "Do you use snus daily?" with the answer alternatives: "Yes" or "No". Participants having answered Yes to any or both questions were categorized as being regular tobacco users.

The AUDIT-C is a three-item alcohol screening tool used to assess harmful consumption of alcohol [24], consisting of the following questions asked at baseline and the 12-month follow-up: "how often do you have a drink of alcohol?" (never, monthly, 2–4 times a month, 2–3 times a week, 4 or more times a week), "how many standard drinks containing alcohol do you have on a typical day?" (1 or 2, 3 or 4, 5 or 6, 7–9, 10 or more), and "how often do you have four (woman), or five (man) or more drinks on one occasion?" (never, less than monthly, monthly, weekly, daily or almost daily). The AUDIT-C generates a score between 0–12.

**Maladaptive coping.** To investigate participants' strategies to cope with different stressors, a Swedish version of the Brief COPE questionnaire was used at baseline and 12 months of follow-up [25]. The questionnaire consists of 28 statements divided into 14 subscales: self-blame, self-distraction, active coping, denial, use of emotional support, use of instrumental support, substance use, behavioral disengagement, venting, positive reframing, planning, humor, acceptance, and religion. Respondents rated each statement on a 4-point Likert scale, ranging from 1 "I haven't been doing this at all" to 4 "I have been doing this a lot" [26]. The 14 subscales are further divided into adaptive and maladaptive coping, with venting, denial, substance use, behavioral disengagement, self-distraction, and self-blame classified as maladaptive with a continuous total score ranging from 12 to 48 [27,28].

**COVID-19-related worries.** Worries relating to the pandemic were asked at 6- and 12-months of follow-up. The COVID-19 Worry Scale (CWS) consists of seven items regarding participants' worry about COVID-19 infection; concerns about themselves, their family, and friends being affected by COVID-19, answered on a 4-point Likert scale ranging from 1 (not at all) to 4 (very much), with a total score ranging from 7 to 28 [29]. The CWS has demonstrated adequate psychometric properties [30,31]. In addition to the CWS, two additional questions were asked regarding potential worries experienced by clinically active manual therapists amidst the COVID-19 pandemic: "How worried have you been of infecting patients/clients?" and "Have you experienced worries related to your clinical practice as a chiropractor/naprapath with relation to the COVID-19 pandemic?" with the same answer alternatives. Further, a free-text question was also added, asking participants to list the three largest worries related to their clinical practice and the COVID-19 pandemic with 6 months recall time: "List the three main worries related to your clinical practice and the COVID-19 pandemic". In the beginning of the section regarding COVID-19-related worries, it was specified that the questions related to the past six months.

**Subjective health impact by the COVID-19 pandemic.** Participants were asked about their subjective physical and mental health and the COVID-19 pandemic at baseline and after 12 months of follow-up: "Has your physical health been impacted by the COVID-19 pandemic?" and "Has your mental health been impacted by the COVID-19 pandemic?" with the answer alternatives "Yes, very negative", "Yes, negative", "No, not at all", "Yes, positive", "Yes, very positive". Those

answering "Yes", in either negative or positive direction received a separate follow-up question on how their impacted physical and/or mental health changed the possibility to carry out clinical work, with the same answer alternatives.

Additionally, those answering "Yes" in either positive or negative direction to the questions described above had the possibility to deepen their answer in free text. At baseline, the questions were: "Comments regarding how the COVID-19 impacted your physical health", "Comments regarding how the COVID-19 impacted your mental health", "Comments regarding how your impacted physical health due to the COVID-19 pandemic affected your work ability", and "Comments regarding how your impacted mental health due to the COVID-19 pandemic affected your work ability". At the 12-month follow-up, the questions were: "Comments regarding how the COVID-19 impacted your physical health the last six months", "Comments regarding how the COVID-19 impacted your mental health the last six months", "Comments regarding how your impacted physical health due to the COVID-19 pandemic affected your work ability the last six months", and "Comments regarding how your impacted mental health due to the COVID-19 pandemic affected your work ability the last six months".

**Free text variables.** Participants had the opportunity to express their experiences freely in several free text questions and statements concerning their health, lifestyle and the COVID-19 pandemic. Apart from those regarding COVID-19-related worries and subjective health impact by the COVID-19 pandemic described above, participants were asked the following question at baseline: "What have you done to promote your own health during the COVID-19 pandemic?", and "What have you done to promote your own health the past six months?" at the 12-month follow-up.

## Statistical analysis

Descriptive statistics were computed for participant baseline characteristics with continuous variables presented as mean and standard deviation (SD), and categorical variables as frequencies and percentages. Generalized Estimating Equations (GEE) with exchangeable correlation matrix was used to study changes in the sample average for physical activity, sedentary time, maladaptive coping strategies, COVID-19-related worries, impaired sleep, tobacco and alcohol use, and subjective health impact by the COVID-19 pandemic over time. Separate models were conducted for each variable over the follow-up period, and stratified analyses were conducted based on sex (male/female), median age (≤ 34 years/ >34 years), and business constellation (working alone/with few or with many colleagues) with the addition of an interaction term between group and time to study the change in the trajectory slope over time between the groups. Statistical analyses were carried out in R version 4.1.4, GEE was conducted with the 'geepack' package [32] for continuous and nominal scale data, and the 'multgee' package [33] was used for ordinal data. Furthermore, a qualitative content analysis was used to code and categorize participants' data from free-text answers [34]. The answers were read through twice by two of the authors (NW, IA) to get an overview of the content. The text was thereafter condensed into meaning units and later coded based on discussion within the research group. Subcategories, categories, and themes were formed, and counts of each subcategory were calculated.

## Results

A total of 850 participants of 1718 eligible agreed to participate in the study. Thirty-four participants were excluded due to not providing information regarding one of the eligibility criteria (being clinically active or not), resulting in a study sample of 816, and a response proportion of 47%. Over the course of the study, 36 participants withdrew their participation, and the response rate at the follow-up surveys was around 80% (Fig 1).

Number of participants invited, agreeing to participate, excluded or withdrawing from the study, and answering at each time-point.

Baseline characteristics of the study sample are presented in Table 1. The mean age of the sample was 44 years, and 46% were women. The proportion of naprapaths and chiropractors amounted to 68% and 32% respectively. In total, 22% of the participants reported working alone.

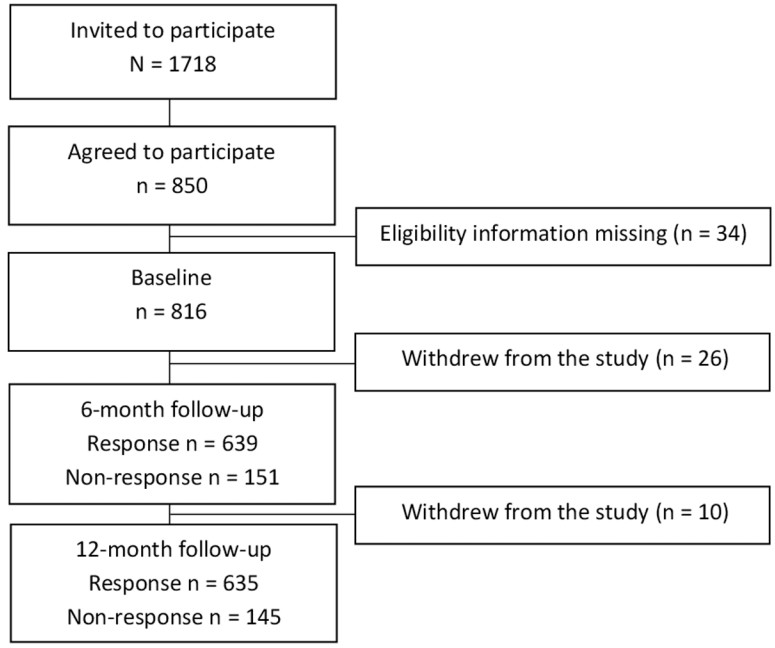

**Fig 1. Study flow chart.**

**Table 1. Baseline characteristics of study participants.**

| Variable | nᵃ | Total |
|---|---|---|
| Age, mean (SD) | 816 | 44 (11.2) |
| **Sex, n (%)** | 816 | |
| Male | | 441 (54) |
| Female | | 375 (46) |
| **Occupation, n (%)** | 816 | |
| Licensed Naprapath | | 508 (62) |
| Licensed Chiropractor | | 257 (31) |
| Naprapath undergoing licensing practice | | 46 (6) |
| Chiropractor undergoing licensing practice | | 5 (1) |
| Number of hours clinically active/week previous 3 months, mean (SD) | 809 | 28.3 (11.4) |
| Other employment, n (%) | 813 | 175 (22) |
| Business owner, n (%) | 746 | 557 (75) |
| Business constellation (working alone), n (%) | 810 | 175 (22) |

ᵃNumber of participants answering the question. SD, standard deviation.

## Longitudinal lifestyle patterns

Longitudinal lifestyle behaviors among the participants during a year are presented in Tables 2–4, for the whole sample, and stratified by sex, median age, and business constellation.

The change between baseline and the months follow-up in physical exercise, total physical activity, and sedentary time during the year for the whole study sample was −5 minutes (95% CI: −7.9, −2.6), −12 minutes (95% CI: −17.6, −5.5), and 0.2 hours (95% CI: 0.0, 0.4), respectively, see Table 2. Those working alone increased their physical exercise by

**Table 2. Physical activity and sedentary time at baseline and the 12 months follow-up in all and stratified by sex, age, and business constellation.**

| | All | Women | Men | ≤ 34 years | > 34 years | Working alone | Not working alone |
|---|---|---|---|---|---|---|---|
| **Physical exercise[a]** | | | | | | | |
| Mean min/week at baseline (95% CI) | 89.8 (87.0, 92.6) | 85.9 (81.6, 90.1) | 93.2 (89.5, 96.8) | 93.0 (89.2, 96.8) | 86.8 (82.7, 90.8) | 86.5 (81.6, 91.5) | 91.7 (88.4, 95.1) |
| Mean min/week at 12 months (95% CI) | 84.5 (81.4, 87,6) | 81.3 (76.7, 85.9) | 87.3 (83.2, 91.5) | 85.5 (81.0, 90.0) | 83.5 (79.2, 87.8) | 83.0 (77.7, 88.3) | 85.6 (81.7, 89.4) |
| *Mean difference in physical exercise baseline to 12 months (95% CI)* | *−5.3 (−7.9, −2.6)* | *1.27 (−4.1, 6.6)* | | *−4.3 (−9.5, 1.0)* | | *6.5 (0.1, 13.0)* | |
| **Everyday physical activity[b]** | | | | | | | |
| Mean min/week at baseline (95% CI) | 98 (95.7, 100.3) | 101.0 (97.9, 104.1) | 95.4 (92.2, 98,7) | 100.5 (97.6, 103,5) | 95.3 (91.8, 98,7) | 100.2 (96.6, 103.9) | 96.8 (93.9, 99.7) |
| Mean min/week at 12 months (95% CI) | 96.9 (94.4, 99,4) | 98.7 (95.2, 102.1) | 95.4 (91.9, 99.0) | 99.0 (95.8, 102.2) | 94.6 (90.6, 98.5) | 96.1 (91.8, 100.3) | 97.7 (94.6, 100.7) |
| *Mean difference in everday physical activity baseline to 12 months (95% CI)* | *−1.05 (−3.5, 1.4)* | *−2.3 (−7.3, 2.7)* | | *0.9 (−4.2, 5.9)* | | *−4.6 (−10.8, 1.6)* | |
| **Total physical activity[c]** | | | | | | | |
| Mean min/week at baseline (95% CI) | 277.6 (271.2, 283.9) | 273 (263, 282) | 282 (273, 290) | 281 (272, 290) | 274 (265, 283) | 273 (262, 284) | 280 (272, 288) |
| Mean min/week at 12 months (95% CI) | 266.0 (258.8, 273.2) | 261 (251, 272) | 270 (260, 280) | 266 (255, 276) | 266 (256, 276) | 262 (250, 274) | 269 (260, 278) |
| *Mean difference in total physical activity baseline to 12 months (95% CI)* | *−11.5 (−17.6, −5.5)* | *0.2 (−12.1, 12.5)* | | *−7.5 (−19.8, 4.8)* | | *8.6 (−5.3, 22.4)* | |
| **Sedentary time[d]** | | | | | | | |
| Mean hours/day at baseline (95% CI) | 4.3 (4.1, 4.4) | 4.1 (3.9, 4.4) | 4.4 (4.1, 4.6) | 4.6 (4.4, 4.8) | 3.9 (3.7, 4.2) | 4.0 (3.8, 4.3) | 4.4 (4.2, 4.6) |
| Mean hours/day at 12 months (95% CI) | 4.6 (4.3, 4.6) | 4.3 (4.0, 4.5) | 4.6 (4.3, 4.9) | 4.9 (4.6, 5.2) | 4.1 (3.8, 4.3) | 4.1 (3.8, 4.4) | 4.7 (4.4, 4.9) |
| *Mean difference in sedentary time baseline to 12 months (95% CI)* | *0.2 (0.0, 0.4)* | *−0.1 (−0.4, 0.3)* | | *0.2 (−0.2, 0.5)* | | *−0.2 (−0.6, 0.1)* | |

95% CI, 95% confidence interval.

[a]Minutes of physical exercise in a normal week (e.g., running, fitness training, ball sports).

[b]Minutes of everyday physical activity in a normal week (e.g., walking, cycling, gardening).

[c]Sum of minutes of exercise per week multiplied by 2 (to account for the higher intensity of exercise) and minutes of daily activity per week.

[d]Number of hours per day spent sedentary.

7 minutes (95% CI: 0.1, 13.0) compared to those not working alone, however, no differences in the trajectory slope of change between the other groups in the separate strata were observed.

The use of maladaptive coping strategies reduced during the follow-up period (−0.3, 95% CI: −0.6, −0.0), however, there were no differences between groups. COVID-19-related worries also reduced during the follow-up period (−0.7, 95% CI: −0.9, −0.5). So did worry about infecting patients (−0.1, 95% CI: −0.2, −0.0), and worry about clinical practice (−0.1,

**Table 3. Psychological factors, impaired sleep, and subjective health impact by the pandemic at baseline and the 12-months follow-up in all and stratified by sex, age, and business constellation.**

| | All | Women | Men | ≤ 34 years | > 34 years | Working alone | Not working alone |
|---|---|---|---|---|---|---|---|
| **Maladaptive coping[a]** | | | | | | | |
| Mean score at baseline (95% CI) | 21.5 (21.2, 21.9) | 22.7 (22.3, 23.1) | 20.5 (20.1, 20.9) | 21.8 (21.4, 22.2) | 21.3 (20.8, 21.8) | 21.1 (20.5, 21.6) | 21.8 (21.4, 22.1) |
| Mean score at 12 months (95% CI) | 21.2 (20.9, 21.6) | 22.4 (21.9, 22.8) | 20.3 (19.8, 20.7) | 21.5 (21.0, 21.9 | 21.0 (20.5, 21.5) | 20.9 (20.3, 21.4) | 21.4 (21.0, 21.8) |
| *Mean difference in maladaptive coping from baseline to 12 months (95% CI)* | *−0.3 (−0.6, −0.0)* | *−0.1 (−0.7, 0.4)* | | *−0.1 (−0.6, 0.6)* | | *−0.0 (−0.8, 0.7)* | |
| **COVID-19 worry scale[b]** | | | | | | | |
| Mean score at 6 months (95% CI) | 14 (13.6, 14.3) | 14.6 (14.2, 15.1) | 13.4 (12.9, 13.9) | 14.2 (13.7, 14.6) | 13.8 (13.3, 14.3) | 14.3 (13.7, 14.9) | 13.8 (13.4, 14.2) |
| Mean score 12 months (95% CI) | 13.3 (12.9, 13.6) | 14.0 (13.6, 14.5) | 12.6 (12.1, 13.0) | 13.2 (12.8, 13.7) | 13.2 (12.8, 13.7) | 13.7 (13.1, 14.3) | 13.0 (12.6, 13.4) |
| *Mean difference in COVID-19 worry scale from baseline to 12 months (95% CI)* | *−0.7 (−0.9, −0.5)* | *0.3 (−0.2, 0.8)* | | *−0.4 (−0.9, 0.1)* | | *0.1 (−0.5, 0.6)* | |
| **Worry about infecting patients[c]** | | | | | | | |
| Mean score at 6 months (95% CI) | 2.3 (2.3, 2.4) | 2.6 (2.5, 2.6) | 2.1 (2.1, 2.2) | 2.4 (2.3, 2.5) | 2.3 (2.2, 2.4) | 2.3 (2.3, 2.4) | 2.3 (2.2, 2.4) |
| Mean score at 12 months (95% CI) | 2.3 (2.2, 2.3) | 2.5 (2.4, 2.6) | 2.1 (2.0, 2.1) | 2.3 (2.2, 2.4) | 2.2 (2.1, 2.3) | 2.3 (2.1, 2.4) | 2.2 (2.2, 2.3) |
| *Mean difference in worry about infecting patients from 6 months to 12 months (95% CI)* | *−0.1 (−0.2, −0.0)* | *0.0 (−0.1, 0.1)* | | *−0.0 (−0.2, 0.1)* | | *0.1 (−0.1, 0.2)* | |
| **Worry about clinical practice[d]** | | | | | | | |
| Mean score at 6 months (95% CI) | 2.3 (2.2, 2.3) | 2.5 (2.4, 2.6) | 2.1 (2.0, 2.2) | 2.3 (2.2, 2.4) | 2.2 (2.1, 2.3) | 2.3 (2.1, 2.4) | 2.2 (2.2, 2.3) |
| Mean score at 12 months (95% CI) | 2.1 (2.1, 2.2) | 2.3 (2.2, 2.4) | 2.0 (1.9, 2.1) | 2.2 (2.1, 2.3) | 2.1 (2.0, 2.2) | 2.2 (2.1, 2.3) | 2.1 (2.0, 2.2) |
| *Mean difference in worry about clinical practice from 6 months to 12 months (95% CI)* | *−0.1 (−0.2, −0.0)* | *−0.1 (−0.2, 0.1)* | | *0.0 (−0.1, 0.2)* | | *0.0 (−0.2, 0.2)* | |
| **Impaired sleep[e]** | | | | | | | |
| Proportion at baseline (95% CI) | 0.09 (0.07, 0.11) | 0.11 (0.08, 0.14) | 0.07 (0.04, 0.09) | 0.08 (0.06, 0.11) | 0.09 (0.06, 0.12) | 0.12 (0.07, 0.17) | 0.08 (0.06, 0.01) |
| Proportion at 12 months (95% CI) | 0.10 (0.08, 0.12) | 0.12 (0.08, 0.15) | 0.09 (0.06, 0.11) | 0.10 (0.7, 0.14) | 0.10 (0.07, 0.13) | 0.11 (0.06, 0.16) | 0.10 (0.07, 0.12 |
| *Mean difference in proportion of impaired sleep from baseline to 12 months (95% CI)* | *0.01 (−0.01, 0.04)* | *−0.01 (−0.06, 0.04)* | | *0.00 (−0.04, 0.06)* | | *−0.03 (−0.10, 0.04)* | |
| **Subjective physical health impact by the COVID-19 pandemic** | | | | | | | |

*(Continued)*

**Table 3.** (Continued)

| | All | Women | Men | ≤ 34 years | > 34 years | Working alone | Not work-ing alone |
|---|---|---|---|---|---|---|---|
| *Cumulative odds ratio from baseline to 12 months (95% CI)*[f] | 0.9 (0.7, 1.1) | 1.2 (0.8, 1.9) | | 0.9 (0.6, 1.3) | | 0.9 (0.6, 1.4) | |
| **Subjective mental health impact by the COVID-19 pandemic** | | | | | | | |
| *Cumulative odds ratio from baseline to 12 months (95% CI)*[f] | 0.5 (0.4, 0.6) | 1.0 (0.7, 1.5) | | 0.7 (0.5, 1.0) | | 1.0 (0.6, 1.6) | |

95% CI, 95% confidence interval.

[a]Sum score of maladaptive coping strategies of Brief COPE, items: 1, 3, 4, 6, 8, 9, 11, 13, 16, 19, 21, 26. Range 12–48.

[b]Total score ranging 7–28.

[c]Total score ranging 1–4.

[d]Total score ranging 1–4.

[e]Difficulties falling asleep and waking up during the night in combination with daytime interference of activities several times a week or always/every day.

[f]Ordinal variable with categories: "Yes, very negative," "Yes, negative," "No, not at all," "Yes, positive," and "Yes, very positive." Estimates represents the cumulative odds ratios for transitioning one step to a higher category over time (towards "Yes, very positive"), with "Yes, very negative" as the reference category.

95% CI: −0.2, −0.0), as seen in Table 3, without any clear differences between groups in the stratified analyses. There was no change in the proportion of participants with impaired sleep during the follow-up. Furthermore, there were lower odds of better subjective mental health impact by the pandemic (OR: 0.5, 95% CI: 0.4, 0.6), however, there were no differences between groups, see Table 3.

Risky scores of alcohol consumption did not change during the follow-up period in the entire group; however, participants below median age (34 years) reduced their alcohol consumption slightly while those older than median age (34 years) increased their alcohol consumption slightly during follow-up (difference in change: −0.2, 95% CI: −0.4, −0.0). Furthermore, the proportion of women regularly using tobacco decreased compared to males during follow-up (−0.04, 95% CI: −0.07, −0.00), see Table 4.

## Free text answers

All subcategories, categories, and themes emerging from the free text answers are presented in Supporting S1-S12 Tables in S1 File.

We asked: "*What have you done to promote your own health during the COVID-19 pandemic*?" At baseline, 664 individuals answered this question. Three overarching themes emerged: promoting physical health (including physical activity, nutrition, sleep, keeping routines and using alternative methods), promoting mental health (including health promotional activities, using the physical environment and handling stress) and adhering to the official recommendations. At the 12 months follow-up, 486 answers were retrieved, with very similar themes. However, at 12 months minimizing alcohol was mentioned as an action to promote physical health, and changing profession emerged as a new category, see Table 5.

Respondents were asked "*How did the COVID-19 pandemic impact your physical health*?" At baseline, 202 individuals reported themes relating to negative impact (such as worse physical condition and lifestyle, covid infection and other illnesses), changes due to the official recommendations (including change in exercise routines and social interaction), but 70 responses concerned themes relating to positive impact (including less infections and improved lifestyle). The 99 responses at 12 months follow-up gave no additional themes, see Table 6.

**Table 4. Alcohol and tobacco consumption at baseline and the 12 months follow-up in all and stratified by sex, age, and business constellation.**

| | All | Women | Men | ≤ 34 years | > 34 years | Working alone | Not working alone |
|---|---|---|---|---|---|---|---|
| **AUDIT-C[a]** | | | | | | | |
| Mean risk score at baseline (95% CI) | 3.1 (3.0, 3.2) | 2.8 (2.6, 2.9) | 3.4 (2.2, 3.6) | 3.0 (2.9, 3.2) | 3.2 (3.1, 3.4) | 3.2 (3.0, 3.4) | 3.1 (2.9, 3.2) |
| Mean risk score at 12 months (95% CI) | 3.1 (3.0, 3.2) | 2.7, (2.6, 2.8) | 3.4 (3.2, 3.6) | 2.9 (2.7, 3.0) | 3.3 (3.1, 3.4) | 3.1 (2.9, 3.3) | 3.0 (2.9, 3.2) |
| *Mean difference in alcohol risk score from baseline to 12 months (95% CI)* | *−0.0 (−0.1, 0.0)* | *−0.1 (−0.2, 0.1)* | | *−0.2 (−0.4, −0.1)* | | *−0.0 (−0.2, 0.2)* | |
| **Tobacco consumption[b]** | | | | | | | |
| Proportion at baseline months (95% CI) | 0.16 (0.13, 0.19) | 0.8 (0.05, 0.10) | 0.23 (0.19, 0.27) | 0.18 (0.14, 0.22) | 0.14 (0.11 0.17) | 0.15 (0.09, 0.20) | 0.16 (0.13, 0.19) |
| Proportion 12 months (95% CI) | 0.16 (0.13, 0.19) | 0.06 (0.03, 0.08) | 0.25 (0.21, 0.29) | 0.17 (0.13, 0.21) | 0.15 (0.12, 0.19) | 0.16 (0.10, 0.22) | 0.16 (0.13, 0.19) |
| *Mean difference in proportion of tobacco consumption from baseline to 12 months (95% CI)* | *0.00 (−0.02, 0.02)* | *−0.04 (−0.07, −0.00)* | | *−0.03 (−0.07, 0.01)* | | *0.02 (−0.02, 0.06)* | |

95% CI, 95% confidence interval.

[a]Total score ranging 0–12.

[b]Smoking or using snus daily.

Respondents stating that their *physical health was impacted by the COVID-19 pandemic*, were asked "*how did this impact your work ability*". At baseline, 74 individuals responded, with themes such as change in physical capacity and workload (both increased and decreased), as well as consequences such as long-covid, sickness absence and musculo-skeletal pain. At the 12-month follow-up, 23 individuals answered this question with the same themes, see Table 7.

The question "*How did COVID-19 impact your mental health?*" yielded 305 responses at baseline. The resulting themes included mental ill-health, social consequences, worry (about COVID and work), but also 25 positive consequences were reported (such as more time with family). The last theme concerned double standards, where the respondents expressed a conflict between the official recommendations and their inability to follow these when working as manual therapists. At the 12-month follow-up, 135 answers were retrieved, and a new category was included in the mental ill-health theme: illness, see Table 8.

Respondents stating that their *mental health was impacted by the COVID-19 pandemic*, were asked "*how did this impact your work ability*". As for physical health, both negative themes (anxiety, worry, and cognitive effects influencing work) and positive themes (less stress, more energy) consequences were recorded. At baseline, 43 individuals answered this question, and no new themes emerged from the 29 responses at 12 months, see Table 9.

Participants who answered having experienced worries related to their clinical practice as a chiropractor/naprapath in relation to the COVID-19 pandemic to a small/moderate/large degree were asked to list the three main worries related to their clinical practice and the COVID-19 pandemic at the 6-month, and 12-month follow-up. The two themes emerging with categories having the highest count were "economic worry", and "worry about infection" in relation to personal, colleagues', and patients' safety. Furthermore, themes related to existential worry, uncertainty about official recommendations, and a general concern for patients' health also emerged, see Table 10.

**Table 5. What have you done to promote your own health during the COVID-19 pandemic.**

| Baseline (n = 664[a]) | 12-month follow-up (n = 486[a]) | |
|---|---|---|
| Total count = 1645[b] | Total count = 1026[b] | |
| Category, number of counts (%)[c] | Category, number of counts (%)[c] | Theme |
| Engaging in exercise/physical activity, 508 (30.9) | Engaging in exercise/physical activity, 432 (42.1) | Activities promoting physical health |
| Ensuring good nutrition, 259 (15.7) | Ensuring good nutrition/restricting alcohol, 205 (20.0) | |
| Getting enough sleep, 101 (6.1) | Getting enough sleep, 90 (8.8) | |
| Keeping routines, 22 (1.3) | Keeping routines, 23 (2.2) | |
| Using alternative methods, 34 (2.1) | Using alternative methods, 9 (0.9) | |
| Engaging in health promotion, 69 (4.2) | Engaging in health promotion, 79 (7.7) | Activities promoting mental health |
| Using the physical environment to my advantage, 209 (12.7) | Using the physical environment to my advantage, 49 (4.8) | |
| Stress handling, 89 (5.4) | Stress handling, 76 (7.4) | |
| | Change profession, 2 (0.2) | |
| Recommendations, 354 (21.5) | Recommendations, 61 (5.9) | Adhering to recommendations |

[a]Number of participants answering the question.

[b]The total number of codes categorized into different subcategories.

[c]Each participant's response could generate several codes and contribute to multiple underlying subcategories forming each category. One category could therefore comprise a higher number of counts than the total number of participants answering the question. The percentage refers to the percentage of times a category was mentioned of the total number of counts.

## Discussion

### Summary of findings

In this study, we found that Swedish manual therapists rated their physical and mental health was impacted by the pandemic. There was a small decrease in physical activity and sedentary time increased slightly as well as subjective mental health impact by the pandemic over one year. Maladaptive coping strategies and COVID-19-related worries decreased during this time. Furthermore, there were small differences in changes over time between younger and older participants in alcohol consumption, and between women and men in tobacco consumption, however, none of the differences observed for coping, worries, as well as tobacco and alcohol consumption were likely clinically relevant.

The free-text answers, that deepened the understanding of what we aimed to study, also indicated no clear changes over time. The manual therapists made considerable effort in health promotion during the COVID-19 pandemic, as they clearly strived to keep physically healthy by addressing important lifestyle factors, physical activity, nutrition and sleep. Also, mental health was addressed using promotional activities such as physical activity, deliberate stress-handling, outdoor stay and social interaction (albeit different than pre-pandemic times).

### Discussion of findings

Several studies of health personnel during the pandemic found a high degree of stress and negative influence on mental health [10–12]. This is not surprising as this was a time of a heavy load on those caring for severely ill patients. Manual therapists in Sweden were not in the same caregiver category, but their work was still challenging as it cannot be carried out without getting close to the patient. We have reported on the mental and musculoskeletal health of this cohort previously, both which were found to be good [16] during the pandemic. Still, many of the issues reported in previous studies,

**Table 6. Comments regarding how the COVID-19 pandemic impacted your physical health.**

| Baseline (n=202[a]) | 12-month follow-up (n=99[a]) | |
|---|---|---|
| Total counts=342[b] | Total counts=147[b] | |
| Category, number of counts (%)[c] | Category, number of counts (%)[c] | Theme |
| Worse physical condition, 111 (32.5) | Worse physical condition, 33 (22.4) | Negative impact |
| Worse lifestyle, 15 (4.4) | Worse lifestyle, 8 (5.4) | |
| Mental impact, 49 (14.3) | Mental impact, 21 (14.3) | |
| Covid and other infections, 47 (13.7) | Covid and other infections, 34 (23.1) | |
| Illness, 22 (6.4) | Illness, 12 (8.2) | |
| Less infections, 23 (6.7) | Less infections, 5 (3.4) | Positive impact |
| More exercise, 32 (9.4) | More exercise, 15 (10.2) | |
| Improved lifestyle, 15 (4.4) | Improved lifestyle, 6 (4.1) | |
| Changed exercise, 9 (2.6) | Changed exercise, 2 (1.3) | Changes due to official recommendations |
| Social distancing, 8 (2.3) | Social distancing, 4 (2.7) | |
| Less work, 8 (2.3) | Less work, 6 (4.1) | |
| More work, 3 (0.9) | More work, 1 (0.7) | |

[a]Number of participants answering the question.

[b]The total number of codes categorized into different underlying subcategories.

[c]Each participant's response could generate several codes and contribute to multiple underlying subcategories forming each category. One category could therefore comprise a higher number of counts than the total number of participants answering the question. The percentage refers to the percentage of times a category was mentioned of the total number of counts.

**Table 7. Comments regarding how your impacted physical health due to the COVID-19 pandemic affected your work ability.**

| Baseline (n=74[a]) | 12-month follow-up (n=23[a]) | |
|---|---|---|
| Total count=99[b] | Total count=34[b] | |
| Category, number of counts (%)[c] | Category, number of counts (%)[c] | Theme |
| Decreased physical capacity, 22 (22.2) | Decreased physical capacity, 1 (2.9) | Change in physical capacity |
| Improved physical capacity, 13 (13.1) | Improved physical capacity, 4 (11.8) | |
| Increased workload, 14 (14.1) | Increased workload, 13 (38.2) | Change in workload |
| Decreased workload, 16 (16.2) | Decreased workload, 5 (14.7) | |
| Other consequences, 34 (34.3) | Other consequences, 11 (32.4) | Consequences of COVID-19 |

[a]Number of participants answering the question.

[b]The total number of codes categorized into different underlying subcategories.

[c]Each participant's response could generate several codes and contribute to multiple underlying subcategories forming each category. One category could therefore comprise a higher number of counts than the total number of participants answering the question. The percentage refers to the percentage of times a category was mentioned of the total number of counts.

such as stress, anxiety, and worry were also found in this group when we asked the participants to describe how they were affected.

One explanation for these stable and relatively good health outcomes is that our participants maintained good lifestyle habits and seemed conscientious regarding health promotion measures. Exercising, preferably outdoors, eating well, getting enough sleep and keeping in touch with family and friends may have contributed to these stable health outcomes.

**Table 8. Comments regarding how the COVID-19 pandemic impacted your mental health.**

| Baseline (n = 305[a]) | 12-month follow-up (n = 135[a]) | |
|---|---|---|
| Total number of counts = 598[b] | Total number of counts = 205[b] | |
| Category, number of counts (%)[c] | Category, number of counts (%)[c] | Theme |
| Mental ill-health, 207 (34.6) | Mental ill-health, 61 (29.8) Illness, 5 (2.3) | Mental ill-health |
| Social consequences, 63 (10.5) | Social consequences, 20 (9.8) | Social consequences |
| Limitations in leisure time, 12 (2.0) | Limitations in leisure time, 5 (2.3) | |
| Better mental health, 21 (3.5) | Better mental health, 12 (5.9) | Positive consequences |
| More time with family, 4 (0.7) | More time with family, 1 (0.5) | |
| Fear of COVID-19, 101 (16.9) | Fear of COVID-19, 17 (8.3) | COVID-19-related worry |
| General worry, 49 (8.2) | General worry, 13 (6.3) | |
| Influence on work, 11 (1.8) | Influence on work, 16 (7.8) | Work-related worry |
| Worry about work/economy, 124 (20.7) | Worry about work/economy, 40 (19.5) | |
| Conflict between work and official recommendations, 6 (1.0) | Conflict between work and official recommendations, 15 (7.3) | Double standards/moral |

[a]Number of participants answering the question.

[b]The total number of codes categorized into different underlying subcategories.

[c]Each participant's response could generate several codes and contribute to multiple underlying subcategories forming each category. One category could therefore comprise a higher number of counts than the total number of participants answering the question. The percentage refers to the percentage of times a category was mentioned of the total number of counts.

**Table 9. Comments regarding how your impacted mental health due to the COVID-19 pandemic affected your work ability.**

| Baseline (n = 43[a]) | 12-month follow-up (n = 29[a]) | |
|---|---|---|
| Total number of counts = 54[b] | Total number of counts = 50[b] | |
| Category, number of counts (%)[c] | Category, number of counts (%)[c] | Theme |
| Exhaustion/depression, 21 (38.9) | Exhaustion/depression, 25 (50.0) | Negative consequences on mental health |
| Worry/anxiety, 18 (33.3) | Worry/anxiety, 4 (8.0) | |
| Impact on work, 4 (7.4) | Impact on work, 12 (24.0) | |
| Cognitive consequences, 3 (5.6) | Cognitive consequences, 3 (6.0) | |
| Positive consequences, 8 (14.8) | Positive consequences, 6 (12.0) | Positive consequences on mental health |

[a]Number of participants answering the question.

[b]The total number of codes categorized into different underlying subcategories.

[c]Each participant's response could generate several codes and contribute to multiple underlying subcategories forming each category. One category could therefore comprise a higher number of counts than the total number of participants answering the question. The percentage refers to the percentage of times a category was mentioned of the total number of counts.

Another explanation for our findings may be Sweden's approach to not enforcing a lockdown. The participants in the study were clinically active manual therapists in the year of follow-up, and working is a well-known contributor to good health [35]. Furthermore, they were able to exercise outdoors, which was seen as a health-promoting activity, physically and mentally. Many participants also mentioned that they kept to their usual routines as much as possible in order to stay healthy. A parallel finding from Sweden is a study among university students, who also kept healthy lifestyle behaviors through the pandemic [36].

**Table 10. List the three main worries related to your clinical practice and the COVID-19 pandemic.**

| 6-month follow-up (n = 373[a]) | 12-month follow-up (n = 380[a]) | |
|---|---|---|
| Total number of counts = 787[b] | Total number of counts = 902[b] | |
| Category, number of counts (%)[c] | Category, number of counts (%)[c] | Theme |
| Colleague safety, 2 (0.3) | Colleague safety, 13 (1.4) | Worry about infection |
| Patient safety, 22 (2.8) | Patient safety, 22 (2.4) | |
| Personal safety, 22 (2.8) | Personal safety, 44 (4.9) | |
| Worry about infection, 293 (37.2) | Worry about infection, 247 (27.4) | |
| Economic worry, 381 (48.4) | Economic worry, 450 (49.9) | Economic worry |
| Existential worry, 15 (1.9) | Existential worry, 13 (0.3) | Existential worry |
| Uncertainty about recommendations, 28 (3.6) | Uncertainty about recommendations, 77 (8.5) | Uncertainty about recommendations |
| Concern for patients' health, 24 (3.0) | Concern for patients' health, 36 (4.0) | Concern for patients' health |

[a]Number of participants answering the question.

[b]The total number of codes categorized into different underlying subcategories.

[c]Each participant's response could generate several codes and contribute to multiple underlying subcategories forming each category. One category could therefore comprise a higher number of counts than the total number of participants answering the question. The percentage refers to the percentage of times a category was mentioned of the total number of counts.

As the pandemic progressed, a new health issue emerged: worry. Around the world, students [37], minorities [38], as well as health care workers and non-health care essential workers [39] reported being worried in relation to the pandemic. They were worried about their families getting the virus, the safety of vaccines, lockdowns and governmental policies. In our study, we specifically asked manual therapists about worries relating to COVID-19 infection; concerns about themselves, their family, and friends being affected by COVID-19, as well as concerns about infecting patients. As Sweden did not enforce a lockdown, these were thought to be real concerns for a manual therapist seeing patients. However, the participants scored low on the CWS, both at 6- and 12-month follow-up. It may be that by these points in time (May/June 2021 and November/December 2021), having lived through the pandemic for a year and more, they knew what to expect, and were less worried. Several subcategories related to the COVID-19 vaccination emerged from the free-text questions. In the 6-month follow-up, a few of the therapists raised worries concerning being last of the healthcare workers being vaccinated. As time progressed, worries related to vaccine side effects, or the effectiveness of vaccines, feeling pressured to get vaccinated or restrictions imposed if not getting vaccinated emerged in the 12-month follow-up.

### Strengths and weaknesses

The major strength of this study is the large cohort and the prospective design that enabled to study the changes over time. The good response rates at follow-ups (80%) enable us to study such changes with good validity. Wherever possible, valid questionnaires were used, to avoid misclassification. Non-standardized questions and items were piloted by manual therapists prior to distribution of the baseline questionnaire, to ensure comprehensiveness. Further, the mixed method approach allowed for a detailed and deepened understanding of the study aims. Nevertheless, since it is a challenge to measure lifestyle as alcohol consumption, physical activity and sedentary time with questionnaires, we cannot rule out an underestimation of a risk behavior during the follow-up. If such misclassification differs between men and women, and between older and younger participants we might have missed a potential difference between subgroups. Further, if subgroups differed regarding any other factor related to lifestyle, we might wrongly have concluded that group differences were lacking.

Comparisons of the cohort with eligible participants was possible regarding some variables (age, sex, occupation, and place of business), and only a slight difference with regards to sex was found with a larger proportion of our sample being female (46% versus 40%). Thus, we conclude that the sample is representative.

## Conclusion

Swedish manual therapists maintained good lifestyle habits except for a small decrease in physical activity and slight increase in sedentary behavior and subjective mental health impact by the pandemic over time. There were small differences in terms of maladaptive coping, alcohol consumption, and tobacco consumption, however, these differences were not likely clinically relevant. The therapists seemed conscientious regarding health promotion measures during one year of the COVID-19 pandemic.

## Supporting information

**S1 File. Supporting table S1-S12.**
(DOCX)

## Acknowledgments

The authors would like to express gratitude to Sofie Jonsson for project administration and assistance, and the participating manual therapists for their contribution.

## Author contributions

**Conceptualization:** Iben Axén, Nathan Weiss, Eva Skillgate.

**Data curation:** Iben Axén, Nathan Weiss, Eva Skillgate.

**Formal analysis:** Iben Axén, Nathan Weiss, Eva Skillgate.

**Funding acquisition:** Iben Axén, Nathan Weiss, Eva Skillgate.

**Investigation:** Iben Axén, Nathan Weiss, Eva Skillgate.

**Methodology:** Iben Axén, Nathan Weiss, Eva Skillgate.

**Project administration:** Iben Axén, Nathan Weiss, Eva Skillgate.

**Resources:** Iben Axén, Eva Skillgate.

**Supervision:** Iben Axén, Eva Skillgate.

**Writing – original draft:** Iben Axén, Nathan Weiss, Eva Skillgate.

**Writing – review & editing:** Iben Axén, Nathan Weiss, Eva Skillgate.

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
