## [Decision Letter · Decision Letter 0]

10 Jan 2025

PONE-D-24-54978Health aspects and lifestyle of licensed manual therapists during the COVID-19 pandemic in Sweden; the CAMP cohort studyPLOS ONE

Dear Dr. Weiss,

Thank you for submitting your manuscript to PLOS ONE. After careful consideration, we feel that it has merit but does not fully meet PLOS ONE’s publication criteria as it currently stands. Therefore, we invite you to submit a revised version of the manuscript that addresses the points raised during the review process.

The paper is interesting and overall of good quality. Both referees raised several weaknesses that require a thorough editing of the manuscript..

We look forward to receiving your revised manuscript.

Kind regards,

Guglielmo Campus, Ph.D DDS

Academic Editor

PLOS ONE

Journal Requirements:

3. In the online submission form, you indicated that due to ethical restrictions of disclosing personal data, authors have to seek permission to allow us to make the data used in this study available. Data will be available upon request after permission is granted from the Karolinska Institutet’s Ethics Review Board in Stockholm whose contact is kansli@stockholm.epn.se. Inquiries for data access should first be sent to iben.axen@ki.se, who will then contact the ethics board for permission to openly share the data.

5. Please remove all personal information, ensure that the data shared are in accordance with participant consent, and re-upload a fully anonymized data set. 

Reviewers' comments:

Reviewer's Responses to Questions

**Comments to the Author**

1. Is the manuscript technically sound, and do the data support the conclusions?

Reviewer #1: Yes

Reviewer #2: Yes

2. Has the statistical analysis been performed appropriately and rigorously? 

Reviewer #1: Yes

Reviewer #2: Yes

3. Have the authors made all data underlying the findings in their manuscript fully available?

Reviewer #1: No

Reviewer #2: No

4. Is the manuscript presented in an intelligible fashion and written in standard English?

Reviewer #1: Yes

Reviewer #2: Yes

5. Review Comments to the Author

Reviewer #1: I am grateful for the opportunity to contribute to the peer review process of this manuscript, which refers to a longitudinal study in a COVID management context different from the ordinary one of alternating lockdowns.

The authors have presented very clearly the purpose of their study, the methods used and the results obtained. I would like to express my sincere appreciation to them for the work done. Below are some minor comments and some ideas for a deeper understanding of the study results.

1. In the Introduction the authors should add information about vaccination in Sweden. According to Spetz et al., in Sweden, the vaccination program started on 27th December 2020 and by 5th November 2021, 85% of the adult population had received their first dose. Therefore, the baseline data should refer almost entirely to subjects who were not yet vaccinated, while the follow-up data were collected when the vaccine was fully available and, at 12 months, after having also overcome the major uncertainties associated with vaccination. In my opinion, this aspect should also be considered in the discussion.

2. sub-section Physical activity: If I am not mistaken, according to the cited reference [18], the physical activity data should have been managed in a categorical way, on a scale ranging from 3 to 18 instead of the proposed “minutes of activity”. Instead, the authors considered the minutes, converting the categorical response into the central numerical value of the class. It is not clear to me, therefore, whether this method of converting categorical data into a continuous variable (minutes) has had any validation in the literature. As a small comment, regarding the response alternatives: “1-30 minutes”, “30-60 minutes”, “60-90 minutes”, etc., a more accurate data collection would be obtained, for example, by indicating "1-29 minutes", "30-59 minutes" and so on. The original indication "Less than 30 minutes" (as reported in [18]) would be preferable to "1-30", in my opinion.

3. At the end of the Tobacco and alcohol use section there is a typo on AUDIT-C.

4. Regarding the free text variables, how was the question formulated at 12 months? Such as baseline (What have you done to promote your own health during the COVID-19 pandemic?) or, for example, "What have you done to promote your own health during the last 12 months?".

5. Please add appropriate citations for R packages gee and multgee.

6. sub-section Longitudinal lifestyle patterns: I believe that indicating the sedentary time variation even in minutes can help the reader.

7. Table 2: Why was the 34-year threshold taken into consideration? I suggest adding a reason. Furthermore, to assist readers who are not experts in GEE methods, I suggest clarifying what is reported in the line "Mean difference in change baseline to 12 months (95% CI)". The same goes for other similar tables.

8. Table 5: To highlight the appearance of the new category "Change profession", I suggest introducing a decimal instead of reporting 0%. Personally, I was very intrigued by the classification of "Change profession" among the activities for the promotion of mental health. Did the respondents themselves give this indication? Since it seems to be a relevant result, perhaps the authors can spend a few more words on it.

9. Table 6 and related text (lines 233-237): If I understand correctly, all 202 individuals who responded to the baseline indicated at least one negative impact, but 70 of them also indicated positive impacts. Is that right?

10. Tables 5-10 overall: I would like to suggest reviewing the indication of percentages. With reference to table 10, it is really confusing to find indications higher than 100%. Perhaps, these particular cases could be better explained in the legend. Or, the analysis procedure of the individual responses could be illustrated in greater detail in the methods, perhaps with an example (even a fictitious one). Finally, why COVID-19-related worries were not asked at the baseline as well?

References

Spetz, M., Lundberg, L., Nwaru, C., Li, H., Santosa, A., Leach, S., ... & Nyberg, F. (2022). The social patterning of Covid-19 vaccine uptake in older adults: A register-based cross-sectional study in Sweden. The Lancet Regional Health–Europe, 15.

Reviewer #2: Introduction

Lines 58 to 62: The authors discuss how the pandemic impacted the lifestyle and mental health of adolescents worldwide. I suggest revising this section by incorporating references from the scientific literature that highlight the pandemic's impact across all life stages: youth, adults, and the elderly. Alternatively, consider including a reference that focuses on the pandemic's impact on the mental health and lifestyle of healthcare professionals.

Methods

Participants: While the methodological details have already been published and do not need to be repeated in the current manuscript, it would be valuable for readers to know the inclusion and exclusion criteria for participants, such as sex, age group, pre-existing clinical conditions, exclusion of participants with prior illnesses that could interfere with the outcomes, as well as the total number of participants recruited or other criteria applied in the study.

Other Considerations

Based on the manuscript itself (and not the reference to the larger study), the study does not appear to align with the characteristics of a classical cohort study in epidemiology. The authors did not explicitly describe (at least in the text) how participants were selected for exposure versus non-exposure to the factor of interest (lifestyle). I suggest that the authors include these details in the manuscript. Additionally, if the study is methodologically a prospective web survey, I strongly recommend replacing the term "cohort study" with "repeated web surveys in the same population" to more accurately reflect the aim of evaluating lifestyle changes. Furthermore, I suggest specifying that this is a mixed-methods (qualitative-quantitative) study.

Follow-up: Please clarify in the methodology how the authors handled loss to follow-up and whether this loss was selective.

6. PLOS authors have the option to publish the peer review history of their article (what does this mean? ). If published, this will include your full peer review and any attached files.

**Do you want your identity to be public for this peer review?** For information about this choice, including consent withdrawal, please see our Privacy Policy .

Reviewer #1: **Yes: ** Antonella Bodini

Reviewer #2: No

---

## [Author Response · Author response to Decision Letter 1]

6 Mar 2025

Reviewer #1: I am grateful for the opportunity to contribute to the peer review process of this manuscript, which refers to a longitudinal study in a COVID management context different from the ordinary one of alternating lockdowns.

The authors have presented very clearly the purpose of their study, the methods used and the results obtained. I would like to express my sincere appreciation to them for the work done. Below are some minor comments and some ideas for a deeper understanding of the study results.

Response: Thank you for your valuable comments which we believe have improved the quality and readability of the manuscript.

1. In the Introduction the authors should add information about vaccination in Sweden. According to Spetz et al., in Sweden, the vaccination program started on 27th December 2020 and by 5th November 2021, 85% of the adult population had received their first dose. Therefore, the baseline data should refer almost entirely to subjects who were not yet vaccinated, while the follow-up data were collected when the vaccine was fully available and, at 12 months, after having also overcome the major uncertainties associated with vaccination. In my opinion, this aspect should also be considered in the discussion.

Response: Thank you for the valuable suggestion. We have added information regarding the Swedish vaccine program in the introduction, se lines 51-56:

“Sweden’s vaccination program against COVID-19 were initiated in the end of 2020 and was implemented in stages depending on risk status. Elderly people and frontline healthcare workers were among the first groups that received the vaccine, and thereafter it was progressively made available for other groups in the population based on age and pre-existing conditions. By November 2021, 85% of the adult population in Sweden had received their first dose [5].”

Since worry was not measured at baseline, we cannot relate that to the vaccination status in the discussion. Nevertheless, we have also raised opinions related to vaccination from the free-text answers in the discussion, see lines 342-346:

“Several subcategories related to the COVID-19 vaccination emerged from the free-text questions. In the 6-month follow-up, a few of the therapists raised worries concerning being last of the healthcare workers being vaccinated. As time progressed, worries related to vaccine side effects, or the effectiveness of vaccines, feeling pressured to get vaccinated or restrictions imposed if not getting vaccinated emerged in the 12-month follow-up. “

2. sub-section Physical activity: If I am not mistaken, according to the cited reference [18], the physical activity data should have been managed in a categorical way, on a scale ranging from 3 to 18 instead of the proposed “minutes of activity”. Instead, the authors considered the minutes, converting the categorical response into the central numerical value of the class. It is not clear to me, therefore, whether this method of converting categorical data into a continuous variable (minutes) has had any validation in the literature. As a small comment, regarding the response alternatives: “1-30 minutes”, “30-60 minutes”, “60-90 minutes”, etc., a more accurate data collection would be obtained, for example, by indicating "1-29 minutes", "30-59 minutes" and so on. The original indication "Less than 30 minutes" (as reported in [18]) would be preferable to "1-30", in my opinion.

Response: Thank you for raising this comment. We managed the physical activity as proposed, however, instead of forming an index with the total points ranging from 3-18 we replaced each score with the central value of minutes for each category. E.g., 30-60 minutes of physical exercise would equate to 90 minutes in total physical activity (45 minutes * 2), instead of 4 points in the PA-index. This was done to make it easier for the reader to interpret the results.

Further, we agree that the questions would be more accurate as proposed, with categories “1-29 minutes” or less than 30 minutes as originally constructed. However, we do not believe this modification of the response alternatives resulted in a large misclassification of the measure.

3. At the end of the Tobacco and alcohol use section there is a typo on AUDIT-C.

Response: Thank you for noticing this typo. It has been corrected.

4. Regarding the free text variables, how was the question formulated at 12 months? Such as baseline (What have you done to promote your own health during the COVID-19 pandemic?) or, for example, "What have you done to promote your own health during the last 12 months?".

Response: Thank you for the question. We have added a detailed description of each free-text question.

See line 159-163: “Further, a free-text question was also added, asking participants to list the three largest worries related to their clinical practice and the COVID-19 pandemic with six months recall time: “List the three main worries related to your clinical practice and the COVID-19 pandemic”. In the beginning of the section regarding COVID-19-related worries, it was specified that the question related to the last six months.”

And line 172-183: “Additionally, those answering “Yes” in either positive or negative direction to the questions described above had the possibility to deepen their answer in free text. At baseline, the questions were: “Comments regarding how the COVID-19 impacted your physical health”, “Comments regarding how the COVID-19 impacted your mental health”, “Comments regarding how your impacted physical health due to the COVID-19 pandemic affected your work ability”, and “Comments regarding how your impacted mental health due to the COVID-19 pandemic affected your work ability”. At the 12-month follow-up, the questions were: “Comments regarding how the COVID-19 impacted your physical health the last six months”, “Comments regarding how the COVID-19 impacted your mental health the last six months”, “Comments regarding how your impacted physical health due to the COVID-19 pandemic affected your work ability the last six months”, and “Comments regarding how your impacted mental health due to the COVID-19 pandemic affected your work ability the last six months”.”

And line 185-190: “Participants had the opportunity to express their experiences freely in several free text questions and statements concerning their health, lifestyle and the COVID-19 pandemic. Apart from those regarding COVID-19-related worries and subjective health impact by the COVID-19 pandemic described above, participants were asked the following question at baseline: “What have you done to promote your own health during the COVID-19 pandemic?”, and “What have you done to promote your own health the past six months?” at the 12-month follow-up.”

5. Please add appropriate citations for R packages gee and multgee.

Response: We have added citations to the R packages. Please see lines 202-203

6. sub-section Longitudinal lifestyle patterns: I believe that indicating the sedentary time variation even in minutes can help the reader.

Response: Thank you for the comment. Considering the sedentary time was measured in hours we believe it is more appropriate to present it as such, even though it’s not consistent with the reporting metric of physical activity.

7. Table 2: Why was the 34-year threshold taken into consideration? I suggest adding a reason. Furthermore, to assist readers who are not experts in GEE methods, I suggest clarifying what is reported in the line "Mean difference in change baseline to 12 months (95% CI)". The same goes for other similar tables.

Response: Thank you for the comment. The 34-year cut-off is simply the median age of the sample. This is also described in the statistical analysis, see lines 197-201:

“Separate models were conducted for each variable over the follow-up period, and stratified analyses were conducted based on sex (male/female), median age (≤ 34 years/>34 years), and business constellation (working alone/with few or with many colleagues) with the addition of an interaction term between group and time to study the change in the trajectory slope over time between the groups.”

We have revised the “Mean difference in change baseline to 12 months (95% CI)", for clarity, please see revised tables 2-4.

8. Table 5: To highlight the appearance of the new category "Change profession", I suggest introducing a decimal instead of reporting 0%. Personally, I was very intrigued by the classification of "Change profession" among the activities for the promotion of mental health. Did the respondents themselves give this indication? Since it seems to be a relevant result, perhaps the authors can spend a few more words on it.

Response: We have added a decimal to the tables 5-10. We agree that this category was interesting. We used a manifest content analysis which were very “word centered”, thus, the participants’ answers clearly stated that they changed profession to promote their mental health. Considering only 2 participants mentioned this, we believe the category is too small to elaborate on and give room in the manuscript text.

9. Table 6 and related text (lines 233-237): If I understand correctly, all 202 individuals who responded to the baseline indicated at least one negative impact, but 70 of them also indicated positive impacts. Is that right?

Response: Thank you for the comment. We realize the presentation of the results was not clear. We have revised all tables presenting the qualitative analyses (Table 5-10).

As noted in the footnotes of the tables, each participant’s response could generate several codes and contribute to multiple subcategories forming the categories. Thus, this implies that the total number of counted codes are higher than the total number of participant responses. In the revised tables, the percentage for each category is calculated based on the total number of counted codes instead of number of individual responses (as earlier), we believe this makes it clearer for the reader to interpret the tables.

10. Tables 5-10 overall: I would like to suggest reviewing the indication of percentages. With reference to table 10, it is really confusing to find indications higher than 100%. Perhaps, these particular cases could be better explained in the legend. Or, the analysis procedure of the individual responses could be illustrated in greater detail in the methods, perhaps with an example (even a fictitious one).

Response: Thank you for the comment. Please see our previous response regarding this. We hope the revised tables make it clearer.

Finally, why COVID-19-related worries were not asked at the baseline as well?

Response: We did not consider studying COVID-19-related worries specifically when designing the study. However, as the pandemic progressed, and this was flagged as a concern in other studies, we added the CWS with a 6-month recall at both the 6- and 12-month follow-up.

References

Spetz, M., Lundberg, L., Nwaru, C., Li, H., Santosa, A., Leach, S., ... & Nyberg, F. (2022). The social patterning of Covid-19 vaccine uptake in older adults: A register-based cross-sectional study in Sweden. The Lancet Regional Health–Europe, 15.

Reviewer #2:

Introduction

Lines 58 to 62: The authors discuss how the pandemic impacted the lifestyle and mental health of adolescents worldwide. I suggest revising this section by incorporating references from the scientific literature that highlight the pandemic's impact across all life stages: youth, adults, and the elderly. Alternatively, consider including a reference that focuses on the pandemic's impact on the mental health and lifestyle of healthcare professionals.

Response: Thank you for the valuable comment. Reference number 8 (Lopez-Morales et al. 2024) is based on all age groups. We have revised to change the comparison population in Sweden, see lines 65-67:

“Financial instability, fear of infection and the unpredictable situation led to poor mental health in some countries [8], while in Sweden, measures of mental health remained stable over time in a general population sample of adults [9].”

Methods

Participants: While the methodological details have already been published and do not need to be repeated in the current manuscript, it would be valuable for readers to know the inclusion and exclusion criteria for participants, such as sex, age group, pre-existing clinical conditions, exclusion of participants with prior illnesses that could interfere with the outcomes, as well as the total number of participants recruited or other criteria applied in the study.

Response: Thank you for the comment. The inclusion criteria were clinically active chiropractor or naprapath in Sweden, or undergoing licensing practice, which is described on line 87-89:

“Clinically active chiropractors and naprapaths, licensed by the National Board of Health and Welfare in Sweden, and those undergoing licensing practice were invited to participate, and 816 manual therapists were included.”

We did not exclude participants based on any characteristic or demographics mentioned, or based on pre-existing health conditions, however, these were measured and controlled for in analyses.

Other Considerations

Based on the manuscript itself (and not the reference to the larger study), the study does not appear to align with the characteristics of a classical cohort study in epidemiology. The authors did not explicitly describe (at least in the text) how participants were selected for exposure versus non-exposure to the factor of interest (lifestyle). I suggest that the authors include these details in the manuscript. Additionally, if the study is methodologically a prospective web survey, I strongly recommend replacing the term "cohort study" with "repeated web surveys in the same population" to more accurately reflect the aim of evaluating lifestyle changes. Furthermore, I suggest specifying that this is a mixed-methods (qualitative-quantitative) study.

Response: Thank you for the comment. The analyses do not compare exposure status and a specific outcome of interest, rather the change of exposure status over time in generalized estimating equations (GEE). For dichotomized exposures such as impaired sleep and tobacco consumption, there is a clearly described cut-off for classification. For the other exposures, a continuous scale was used, and we assessed the change in score between baseline and the 12—month follow-up. How the variables were collected and categorized are described under “variables” in the methods section, and how they were handled in the analyses under “statistical analysis”.

Furthermore, we agree that it should be clearly stated that the methods of the study were mixed methods. We have revised and incorporated this into the manuscript. See lines 81-83:

“This study was of mixed methods design and was based on the Corona And Manual Professions (CAMP) study, ClinicalTrials register identifier: NCT04834583. The study was approved by the Swedish Ethical Review Authority (Dnr 2020-03836).”

We do however respectfully disagree regarding the classification of the study the article is based on not being a cohort study due to the use of repeated web surveys. The CAMP cohort study is a classical cohort study in our opinion.

Follow-up: Please clarify in the methodology how the authors handled loss to follow-up and whether this loss was selective.

Response: Please see the revised figure 1 for clarity regarding study flow and attrition. Missing data was not handled specifically in the analyses. Luckily, we had a high response rate in the follow-up surveys, and thus did not examine this further. The potential impact of missing on the results of the study are hence minor.

---

## [Decision Letter · Decision Letter 1]

18 Jul 2025

Health aspects and lifestyle of licensed manual therapists during the COVID-19 pandemic in Sweden; the CAMP cohort study

PONE-D-24-54978R1

Dear Dr. Weiss,

We’re pleased to inform you that your manuscript has been judged scientifically suitable for publication and will be formally accepted for publication once it meets all outstanding technical requirements.

Kind regards,

Laura Kelly, PhD

Division Editor

PLOS One

Additional Editor Comments (optional):

Reviewers' comments:

Reviewer's Responses to Questions

**Comments to the Author**

1. If the authors have adequately addressed your comments raised in a previous round of review and you feel that this manuscript is now acceptable for publication, you may indicate that here to bypass the “Comments to the Author” section, enter your conflict of interest statement in the “Confidential to Editor” section, and submit your "Accept" recommendation.

Reviewer #1: All comments have been addressed

Reviewer #3: All comments have been addressed

2. Is the manuscript technically sound, and do the data support the conclusions?

Reviewer #1: Yes

Reviewer #3: Yes

3. Has the statistical analysis been performed appropriately and rigorously? 

Reviewer #1: Yes

Reviewer #3: Yes

4. Have the authors made all data underlying the findings in their manuscript fully available?

Reviewer #1: Yes

Reviewer #3: Yes

5. Is the manuscript presented in an intelligible fashion and written in standard English?

Reviewer #1: Yes

Reviewer #3: Yes

6. Review Comments to the Author

Reviewer #1: (No Response)

Reviewer #3: Further minor comments on the revision,

Line 94 - enter the year after Jan1st assume is 2021

7. PLOS authors have the option to publish the peer review history of their article (what does this mean? ). If published, this will include your full peer review and any attached files.

**Do you want your identity to be public for this peer review?** For information about this choice, including consent withdrawal, please see our Privacy Policy .

Reviewer #1: **Yes: ** Antonella Bodini

Reviewer #3: No

---

## [Editor Report · Acceptance letter]

PONE-D-24-54978R1

PLOS ONE

Dear Dr. Weiss,

I'm pleased to inform you that your manuscript has been deemed suitable for publication in PLOS ONE. Congratulations! Your manuscript is now being handed over to our production team.

Kind regards,

on behalf of

Dr. Laura Hannah Kelly

Staff Editor

PLOS ONE